# 25-Hydroxy-Vitamin D and Risk of Recurrent Stroke: A Dose Response Meta-Analysis

**DOI:** 10.3390/nu15030512

**Published:** 2023-01-18

**Authors:** Anita Vergatti, Veronica Abate, Aquilino Flavio Zarrella, Fiore Manganelli, Stefano Tozza, Rosa Iodice, Gianpaolo De Filippo, Lanfranco D’Elia, Pasquale Strazzullo, Domenico Rendina

**Affiliations:** 1Department of Clinical Medicine and Surgery, University of Naples “Federico II”, 80131 Naples, Italy; 2Department of Neuroscience, Reproductive and Odontostomatology Science, University of Naples “Federico II”, 80131 Naples, Italy; 3Assistance Publique-Hôpitaux de Paris, Hôpital Robert-Debré, Service d′Endocrinologie-Diabétologie, 75019 Paris, France; 4Internal Medicine, University of Naples “Federico II”, 80131 Naples, Italy

**Keywords:** stroke, 25-hydroxy-vitamin D, meta-analysis

## Abstract

Stroke recurrence significantly improves the prognosis quoad vitam et valetudinem of patients with a first ischemic or haemorrhagic stroke. Other than in bone and skeletal metabolism, vitamin D is involved in the pathogenesis of cardiovascular disorders. This meta-analysis was performed to evaluate the relationship between 25OH-vitamin D [25(OH)D] levels at the first stroke and the stroke recurrence. To 31 July 2022, four prospective studies were identified. The potential non-linear relationship was evaluated by modelling 25(OH)D, using restricted cubic splines of 25(OH)D distribution. The pooled estimated risk (and 95% CI) of the recurrence of stroke, comparing the highest and the lowest levels, was assessed using a random-effect model. A non-linear association was found by dose-response analysis. This study found that 25(OH)D levels at the first stroke ≥9.3 ng/mL were associated with a lower risk of stroke recurrence, compared with 25(OH)D levels ≤8.5 ng/mL. In the pooled analysis, higher 25(OH)D levels at the first stroke significantly reduce the risk of stroke recurrence, with a significant heterogeneity among studies. In conclusion, 25(OH)D levels ≤8.5 ng/mL at the first stroke are significantly associated with a higher risk of recurrent stroke.

## 1. Introduction

### 1.1. Stroke: Definition and Epidemiology

A stroke is defined as a neurological deficit caused by an acute focal injury of the central nervous system, in extenso brain, retina, or spinal cord, and secondary to a vascular cause [1]. The large majority of strokes (ranging from 60 to 90%, according to regional stroke epidemiology) are ischemic and are caused by an impaired or absent perfusion through the blood vessels to the brain. From an etio-pathological point of view, an ischemic stroke can be caused by an arterial occlusion or, very rarely, by vein thrombosis. In the first case, ischemic strokes are usually due to arterial occlusion, secondary to cardio-embolic or endothelial disease. In the second case, an ischemic stroke, secondary to venous infarction, is linked to the occlusion of cerebral veins or venous sinuses. The remaining strokes are haemorrhagic and are etio-pathogenetically linked to the rupture of intracerebral or subarachnoid cerebral arteries.

From an epidemiological point of view, a stroke is a very severe and common event, and it actually represents the third cause of disability and the second cause of death worldwide. The absolute number of prevalent and incident cases of stroke registered worldwide during the calendar year 2019 were over 100 million and 12 million, respectively. During the calendar year 2019, the disability adjusted life years (DALYs), secondary to stroke, were over 143 million; over 6.5 million deaths were due to stroke itself [2,3]. In the United States, approximately 800,000 individuals suffer from a stroke each year [4], of whom more than four-fifths are represented by an ischemic stroke and the remaining by a haemorrhagic stroke [3,4]. Atherosclerosis, arterial hypertension, intracerebral vascular abnormalities, diabetes mellitus, atrial fibrillation, high body mass index, obesity, and smoking cigarettes are the most important known risk factors for stroke [4], but the identification of other possible risk factors is still ongoing. In the past two decades, epidemiological data have demonstrated a substantial increase in the annual incidence of stroke; in addition, all the available estimates suggest a notable increase in stroke absolute incidence in the coming years, linked to the progressive increase in age of the world’s population. The substantial advances that have occurred in the acute treatment of ischemic strokes in the last 10 years has, however, guaranteed a significant and constant increase in the absolute number of patients who have survived a first stroke event. These advances acutely account for the availability of thrombolysis and/or mechanical thrombectomy in case of an ischemic stroke, and of an angiography CT scan with embolization in case of a haemorrhagic stroke. In addition, patients need close monitoring and chronic treatment, such as with anti-platelets drug or anticoagulants [1,2].

### 1.2. Stroke Recurrence

The patients that survive a first stroke are at high risk of stroke recurrence. Stoke recurrence is defined as a sudden functional deterioration in neurological status with a decrease in the National Institute of Health Stroke Scale (NIHSS) of four or more, or a new focal neurological deficit of vascular origin that lasts more than 24 h, at least 28 days after the incident event [5]. The meta-analysis performed by Mohan and colleagues, based on 13 studies performed before 2009 and involving 9115 stroke survivors, demonstrates that the pooled cumulative risk of stroke recurrence progressively increases from 3.1% at 30 days to 39.2% at 10 years after the initial stroke [6,7]. These data were substantially confirmed by a successive, more recent, meta-analysis performed by Lin and colleagues, which analysed 37 studies published from 2009 to 2019 and involved more than 1 million subjects [8]. Furthermore, stroke recurrence increases the risk of mortality for stroke and all cardiovascular causes [9]. Given the above, the prevention of stroke recurrence is extremely important in these cases. Thus, a regular medical follow-up of patients who have survived a first stroke is mandatory, in order to prevent the risk of stroke recurrence and death. This has been made possible by old and new blood biomarkers.

### 1.3. Vitamin D System

Vitamin D is the name given to a group of lipid-soluble steroidal hormones and prohormones. The integrity of the vitamin D endocrine system is essential for human health. The large majority of the vitamin D daily need is synthetized in the skin from 7-dehydrocholesterol upon irradiation by ultraviolet B light (wavelength: 290 to 315 nm). Besides solar induced production, it can also be found in certain foods, in the form of ergocalciferol in vegetables and of cholecalciferol in meat. The following activation of vitamin D from the skin and diet happens in the liver and then in the kidney, through a process of hydroxylation on the carbon atom at C1 and C25, respectively. By international convention, the vitamin D status is evaluated measuring the circulating levels of the major and more stable vitamin D circulating metabolite, the 25-hydroxil-vitamin D [25(OH)D or calcifediol]; meanwhile, the more active vitamin D metabolite is represented by 1,25 di-hydroxyl-vitamin D_3_ [1,25(OH)_2_D_3_ or calcitriol], which performs its biological functions by binding to a membrane-bound and cytoplasmic receptor named the vitamin D receptor (VDR), which can be found in almost all human tissues and cells [10]. In the cell nucleus and after the heterodimerization with the retinoid X receptor, the VDR transcribes the target genes located in the deoxyribonucleic acid (DNA), and mediates the genomic effect of the 1,25(OH)_2_D_3_ [11]. In addition, 1,25(OH)_2_D_3_ works on a rapid time scale through non-genomic effects, involving both calcium- and kinase-activated signaling pathways [12]. Using these two mechanisms, the vitamin D performs both osseous and extra osseous actions. The significance of the extra osseous interaction between 1,25(OH)_2_D_3_-VDR is not completely defined, but it appears that vitamin D may cooperate with other regulators in order to operate in the regulation of immune system, in antimicrobial defense, in xenobiotic detoxification, anti-cancer, in the control of insulin secretion and, possibly, in relation to cardiovascular benefits [11]. Indeed, despite the fact that vitamin D was historically known as a key regulator of bone and mineral metabolism, in recent years its possible pathophysiological role in cardio- and cerebro-vascular disorders has been recognized [6,7]. The mechanisms that explain the physio-pathological link occurring between the vitamin D system and cardiometabolic disorders are not fully understood. Low 25(OH)D serum levels increase parathormone (PTH) concentrations, and high PTH levels are a significant risk factor for cardiovascular and cardiometabolic disorders [13,14]. Zhang et al. [15] reported that higher 25(OH)D levels are associated with a reduced risk of atrial fibrillation, which is a significant risk factor for stroke, as above. According to several meta-analyses, low 25(OH)D levels are a significant risk factor for a first stroke occurrence [16,17,18], but few data are available regarding the relationship between 25(OH)D levels at the first stroke event and the risk of stroke recurrence in survivors of a first haemorrhagic or ischemic event [19]. To fill this gap, this meta-analysis, with a dose response analysis, was performed.

## 2. Materials and Methods

### 2.1. Data Source and Literature Search

This meta-analysis was performed according to the Preferred Reporting Items for Systematic Reviews and Meta-Analyses (PRISMA) guidelines [20]. A systematic search for eligible studies was performed by two independent authors (AV and VA) in PubMed, Google Scholar, Cochrane, and Google Book databases on 31 July 2022. Search keywords included “vitamin d”, “stroke”, “cerebrovascular”, and “transient ischemic attack”. All pertinent studies were screened for duplicated articles. No language restriction was applied. All studies in languages different from Italian and English were translated by a professional translator. All selected studies were acquired in full text.

### 2.2. Study Selection

Three authors (D.R., L.D., F.M.) selected the studies included in this meta-analysis. They satisfy the following criteria: (i) observational prospective or retrospective study design; (ii) enrolment of patients with all kinds of first stroke; (iii) the 25(OH)D levels measured at the onset of symptoms related to the first stroke; (iv) outcome measures, including stroke recurrence; (v) follow-up duration ≥ 3 months; and (vi) selected studies must report a multivariable adjusted odds ratio (OR), hazard ratio (HR), or risk ratio (RR) of the outcomes for the specified category of 25(OH)D.

### 2.3. Data Extraction and Quality Assessment

Two authors (AV and VA) independently extracted the data from the selected studies and prepared the database. The extracted data included the following: (a) first author′s last name, (b) publication year, (c) study design, (d) total number of enrolled patients, (e) gender distribution, (f) mean age of patients, (g) follow-up duration, (h) 25(OH)D levels at the first stroke, (i) occurrence of hypertension, (j) type 2 diabetes mellitus, (k) atrial fibrillation, (l) hypercholesterolemia, (m) smoking habits, (n) stroke recurrence, and (o) mortality rate. In the case of any missing data, they were acquired from the corresponding author via mail. The Newcastle–Ottawa Scale (NOS) for cohort′s studies was applied to assess the quality of the selected studies (www.ohri.ca/programs/clinical_epidemiology/oxford.asp, accessed on 15 January 2023). The NOS score ranges from 0 to 8 points, where 0 means the lowest quality level and 8 means the highest quality level. The discrepancies related to the classification of the studies and to the extracted data, if any, were resolved through a discussion between AV and VA with DR.

### 2.4. Statistical Analysis

All data were expressed as absolute numbers for continuous variables and as percentages for dichotomous variables. The assessment for the shape of the association between 25(OH)D levels at the first stroke and the risk of a recurrent stroke was carried out, including estimates adjusted for the greatest number of potential confounders. Restricted cubic splines with three knots at fixed percentiles (10, 50, and 90%) for the distribution of25(OH)D levels were used to explore the possibility of a non-linear relationship. We performed a two-stage dose-response random-effects meta-analysis [21], taking into account the correlation between the estimates across the categories of 25(OH)D levels. Next, we carried out the risk evaluation of the comparison between the highest versus lowest 25(OH)D. Odds ratios (ORs) and hazard risks (HRs) were extracted from the selected publications, and their standard errors (SEs) were calculated from the respective 95% confidence intervals (CIs). The pooled estimated risk (and 95% CI) was assessed using a random-effect model [22].

A specific category for every median 25(OH)D level was attributed to every corresponding estimate. In case the authors did not specify the median or mean 25(OH)D serum level, the midpoint between the upper and the lower boundary was used. In case the lowest category was open-ended, the lower boundary of the same was set to 0. Meanwhile, if the upper boundary of the highest category was left unspecified, we assumed the category to be of the same amplitude as the precedent one. The Cochrane Q test and the I^2^ statistic tests were used to assess the statistical heterogeneity among the selected studies. Funnel plots were constructed and visually assessed for possible publication bias. In addition, formal tests (Egger′s and Begg′s tests) were used to explore a potential publication bias. In the case of significant funnel plot asymmetry, the pooled estimate was recalculated based on the estimated number of “missing” studies via the “trim and fill” method. All statistical analyses were performed by L.D. using the Stata Corp. software (version 11.2; College Station, TX, USA).

## 3. Results

As reported in Figure 1, after the database search, 71 publications were identified and screened. After the exclusion of studies not meeting the pre-established inclusion criteria, four prospective studies [5,23,24,25], all published between 2016 and 2022, were included in this meta-analysis.

The studies by Qiu and colleagues [23], by Huang and colleagues [24], and by Ji and colleagues [25] enrolled patients suffering from an ischemic stroke only, whereas the study by Li and colleagues enrolled patients with both ischemic and haemorrhagic events [5]. As reported in Table 1, this meta-analysis involves 7717 patients with 496 recurrent events documented during the follow-up [5,23,24,25]. Only one study clarifies whether the recurrent stroke was ischemic or haemorrhagic [5]. In particular, the study by Li and colleagues [5] counts 250 ischemic recurrent strokes, 87 haemorrhagic recurrent strokes and 51 unspecified recurrent strokes. The studies by Qiu and colleagues [23], by Huang and colleagues [24], and by Ji and colleagues [25] were carried out in China [23,24,25], whereas the study by Li and colleagues was performed by collecting data from the United Kingdom Biobank [5]. All studies have a prospective observational design.

Male and female participants were recruited by all studies, with a mean age range of 60.6 to 68 years. The follow-up time ranged from 3 to 86 months. The percentage of participants with hypertension ranged from 54.2% to 80.1%; the percentage of participants with diabetes was between 14.7% and 43.1%; and the percentage of participants with atrial fibrillation ranged between 7.8% and 26.1%. The percentage of smokers was between 16.2% and 26.4%. The evaluation of the “risk of bias” indicated that all studies were low-risk.

First, we evaluated the predictive role of 25(OH)D levels in stroke recurrence, comparing the highest and the lowest percentiles. In the pooled analysis, higher 25(OH)D levels at the first stroke event were associated with a significantly lower risk of stroke recurrence (estimated risk: 0.26; 95% CI: 0.10 to 0.67; *p* = 0.005). A significant heterogeneity among studies was found (*p* < 0.001, I^2^ = 93%) (Figure 2).

Study: first author and year of the related study publication. Comparison: level of 25 hydroxy-vitamin D used for comparison. OR: Odds Ratio. HR: Hazard Ratio. 95% CI: 95% Confidence Intervals. The squares indicate the study-specific risk estimates. The square size quantifies the study-specific statistical weight; horizontal lines indicate 95% CI; diamond indicates the overall risk with its 95% CI. Weights are from random effects analysis.

The dose-response evaluation of the association between 25(OH)D levels and the risk of stroke recurrence was performed, including the prospective studies performed by Qiu and colleagues [23], by Li and colleagues [5], and by Huang and colleagues [24]: overall, they include 7440 participants with 465 suffering a recurrent stroke. The study performed by Ji and colleagues [25] was excluded from the dose-response analysis because patients were dichotomized according to the 25(OH)D values above or below the lowest interquartile (12.7 ng/mL).

As depicted in Figure 3, the dose-response analysis demonstrates a non-linear association between the 25(OH)D levels and the risk of recurrent stroke (*p* for non-linearity < 0.0001). In particular, 25(OH)D levels ≥ 9.3 ng/mL were associated with a significantly lower risk of stroke recurrence, compared with 25(OH)D ≤ 8.5 ng/mL. The estimated lowest risk was observed at 25(OH)D levels equal to 28.1 ng/mL (estimated risk: 0.57, 95% CI: 0.43 to 0.77).

Here, 25(OH)D: vitamin D serum level. The 25OHD was modelled with restricted cubic splines in a multivariate random-effects dose–response model (solid line). Dashed lines represent the 95% confidence intervals for the spline model.

With regard to the identification of a publication′s bias, a visual analysis of the funnel plot indicated asymmetry (Figure 4), but this was not confirmed by the formal tests (Egger: *p* = 0.24, Begg: *p* = 0.31). The “trim and fill” method did not identify any potentially missing study.

## 4. Discussion

The meta-analysis results demonstrated a progressive inverse relationship between 25(OH)D levels and the risk of a recurrent stroke in patients surviving a first haemorrhagic or ischemic event.

Historically, the “canonical” biological properties of the vitamin D system are physiologically linked to the control of calcium-phosphate homeostasis [26]. In effect, interacting with PTH and Fibroblast Growth Factor 23 (FGF23), the vitamin D system regulates the functional interaction between the kidneys, the skeleton, the parathyroid glands, and the gut [26]. Indeed, vitamin D features a negative feedback on PTH and a positive one on FGF23, fitting into a tri-axis.

Since 1936, however, several studies have demonstrated the additional “non-canonical” biological properties of the vitamin D system [27,28]. Normal 25(OH)D levels not only prevent hyperparathyroidism [29], involved in many cardiovascular diseases, but can also regulate many hormonal systems. Renin secretion in the kidney is downregulated by vitamin D signalling, and in this way, it suppresses the renin–angiotensin–aldosterone system, controlling the arterial blood pressure [30,31]. In the peripheral tissue, in particular in muscle, vitamin D improves the insulin sensitivity, improving glucose cellular metabolism and in doing so, reducing the glucose-induced oxidative stress [32,33]. In addition, recently, an association between low 25(OH)D levels and vascular calcifications in humans has been found, probably related to the role of vitamin D in osteoblast differentiation in the vessel wall. Vascular calcification is being associated with a reduced elastance and the compromised structural integrity of arterial vessels [34]. Of interest, clinical evidence demonstrates that low vitamin D levels are associated with dyslipidemia and with the progression of the atherosclerotic process. Experimental data also proved that the macrophages lacking the VDR present an increased uptake of cholesterol, causing the formation and progression of atherosclerotic plaque [35].

Furthermore, the vitamin D system modulates the immune response and the cytokine biosynthesis [36]. In this way, vitamin D interacts with both the innate and adaptive immune system through the binding of the VDR onto immunity cells. This action reduces inflammatory and autoimmunity responses.

Nowadays, the cardiovascular system is considered to be a non-canonical target for the vitamin D system; this is because the VDR and the 1-alpha hydroxylase enzyme are structurally expressed in endothelial and vascular smooth muscle cells and cardiomyocytes, as well as in macrophages and T-cells [37]. In this way, vitamin D modulates the calcium cell influx, regulating the strength and contractility of vascular smooth muscle cells. In addition, it appears to play a role in the modulation of endothelial cell survival and autophagy [38]. This results in Vitamin D regulation in cell differentiation and growth, and in a modulation in the inflammatory response; this is involved in the pathogenesis of atherosclerosis, endothelial dysfunction, damage to the blood–brain barrier, and in the control of other hormonal systems [28,39,40,41,42].

Vitamin D also plays a certain role in reduced drug efficacy in the condition of hypovitaminosis. Drugs that interfere with the adenosine diphosphate (ADP)-mediated platelet aggregation are a first line choice for stroke treatment and for the prevention of recurrence [43]. Lower vitamin D levels are also associated with an inadequate inhibition of ADP-mediated platelet aggregation by clopidogrel and ticagrelor, leaving the patients vulnerable to endothelial event recurrence, as in the case of myocardial infarct and stroke [43]. Indeed, a large cohort study proved that male patients with low vitamin D levels and undergoing elective coronary angiography presented with a multi-vessel involvement and with more severe coronary artery stenosis [44].

Another connection between Vitamin D and cardiovascular health can be found in the ubiquitous expression of both the VDR and the 1-alfa-hydroxylase enzyme. This can be found in several cell types, including cardiomyocytes, endothelial cells, pericytes, and smooth-muscular cells [45,46]. In addition, VDR gene polymorphisms have been linked to stent restenosis following percutaneous coronary intervention [47]. One can assume that this can also play a role in the physio-pathological mechanism of the recurrence of strokes.

In addition, a study by Hagström and colleagues reported a statistically significant association between higher PTH concentrations and cardiovascular death among older Swedish men, also suggesting that a high PTH may be associated with cardiovascular events [48].

For all these reasons, the role of vitamin D in the pathogenesis of cardiometabolic disorders, including stroke, is proposed [49,50]: vitamin D deficiency is finally considered to be a significant risk factor for chronic degenerative and cardiometabolic disorders [51].

Several studies demonstrated a significant relationship between low 25(OH)D levels and mortality for all-cause and for cardiovascular disease [31,52]. In this scenario, however, the VITAL study results demonstrate that the indiscriminate supplementation of vitamin D is not useful for the primary prevention of cardiovascular diseases, although it significantly improves the prognosis quoad vitam et valetudinem of patients with cancer [53]. In this regard, it should be highlighted that most of the participants in large vitamin D randomized controlled trials have high 25(OH)D levels and that the baseline 25(OH)D is not taken into account [54,55].

Our dose-response analysis for the first time establishes a non-linear association between 25(OH)D levels and the recurrence of a stroke, with an unfavourable association between low 25(OH)D levels and the risk of recurrent events in patients surviving a first stroke. In particular, patients who survived a stroke with 25(OH)D levels equal or higher than 9.3 ng/mL at the time of the first stroke have a significantly lower risk of stroke recurrence, compared to patients with very low levels. Recently, there has been a large debate involving several scientific societies about the levels of 25(OH)D and their indication of a deficiency state [10,56,57,58].

Nowadays, 25(OH)D levels lower than 10 ng/mL are recognized by all international scientific societies as insufficient and requiring specific treatment to guarantee an optimal calcium intake and normal PTH serum levels [59]. Conversely, there is reasonable evidence that 25(OH)D levels equal or higher than 30 ng/mL reduce the risk of several, multifactorial, and complex diseases [59]. Nevertheless, our results are supported by genetic mendelian randomized studies [60].

A growing body of experimental and clinical evidence indicates that the vitamin D biological system plays a pivotal role in brain development during infancy and adolescence, and in the maintenance of brain functions in adulthood [12]. In addition, vitamin D plays a neuroprotective role against age-related brain degeneration. Experimental data indicate that both 25(OH)D and 1,25(OH)_2_D_3_ cross the blood–brain barrier. In addition, 1 alfa hydroxylase and Cytochrome P450 Family 24 Subfamily A Member 1 (CYP24A1), a key enzyme for vitamin D catabolism, are expressed in the neurons and glia cells of the human fetal and adult brain [12]. These observations strongly suggest that 1,25(OH)_2_D_3_ can be also synthetized and metabolized locally in the central nervous system. Both the genomic and non-genomic actions of vitamin D are likely to influence brain development, function and maintenance. The VDR is widely distributed in different zones of the human brain, and mediates the genomic and non-genomic actions of vitamin D. In the developing brain, vitamin D, via its genomic actions, regulates neural stem cell proliferation and differentiation, dopaminergic neuron development, and oligodendrocyte differentiation [12]. In the adult brain, always via its genomic actions, vitamin D modulates both the hippocampal neurogenesis and the release of neurotransmitters, such as the inhibitory neurotransmitter gamma-aminobutyric acid (GABA) and dopamine. In the aged brain, the neuroprotective role of vitamin D is linked to the increased expression of multiple neurotrophic factors, such as the nerve growth factor, the glia-derived neurotrophic factor, and neurotrophin-3 and neurotrophin-4. Furthermore, vitamin D is also a regulator of age-related inflammation, reducing the production of interleukin-2, interleukin-10, interleukin-12, and interleukin-1b [12].

In the end, one can speculate that vitamin D is essential to the cellular brain survival.

### Study Strengths and Limitations

Our meta-analysis suffers from a major limitation: there are few studies and, moreover, these are heterogeneous. The heterogeneity of the selected studies is caused by differences in the follow-up duration, the different cut-off of 25(OH)D levels and the different definitions of stroke type or outcome. Furthermore, the studies selected for this meta-analysis consider only 25(OH)D levels, without considering nutritional supplementation and solar ultra-violet B exposure, outperformed by genetic mendelian randomized studies [46,52,60,61]. In turn, the heterogeneity is tempered because three of the selected studies were conducted in the same country (China), and only one enrolled patients from the United Kingdom.

On the other hand, the strict selection criteria and dose response analysis are our major strengths. In particular, the dose response analysis is functional to partially overcome the study limitations and to provide new and intriguing data. In this way, we provided clinicians with a new and additional tool for the better follow-up of patients that had already had a first stroke event.

## 5. Conclusions

In conclusion, the results of this meta-analysis demonstrate that stroke patients with low 25(OH)D levels at the first event have an increased risk of stroke recurrence. In addition, the results of the dose-response analysis demonstrate that the risk of stroke recurrence decreases progressively with an increase in 25(OH)D levels: the estimated highest and lowest risk were observed at 25(OH)D levels lower or equal to 8.5 ng/mL and equal to 28.1 ng/mL, respectively. As observational studies do not allow the establishment of cause–effect relationships, our results will need further randomized controlled trials in order to establish the cut-off 25(OH)D levels in patients that have already experienced a first stroke. In addition, future prospective studies will help to define the independent role of vitamin D in the management of patients suffering from several unrelated conditions, that cooperate in the recurrence of a stroke.

## Figures and Tables

**Figure 1 nutrients-15-00512-f001:**
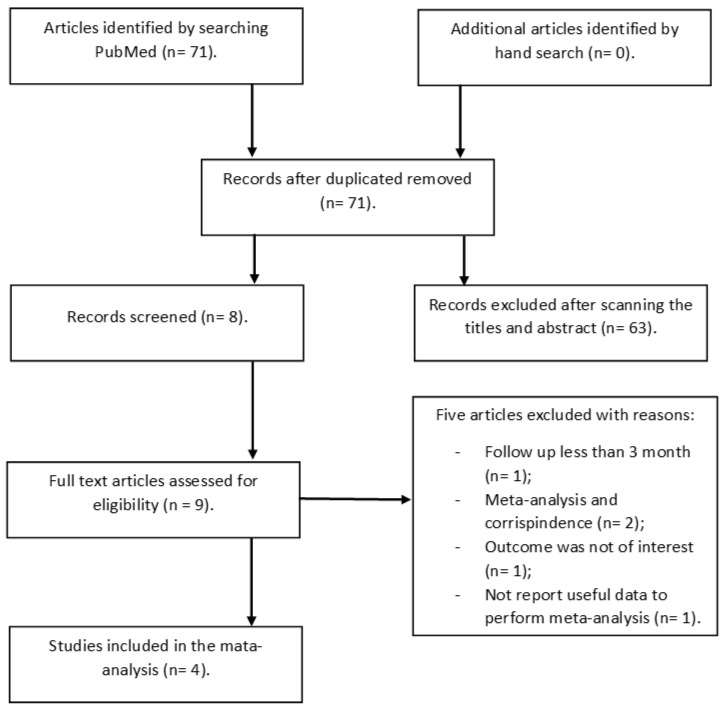
Stepwise procedure for selection of the studies.

**Figure 2 nutrients-15-00512-f002:**
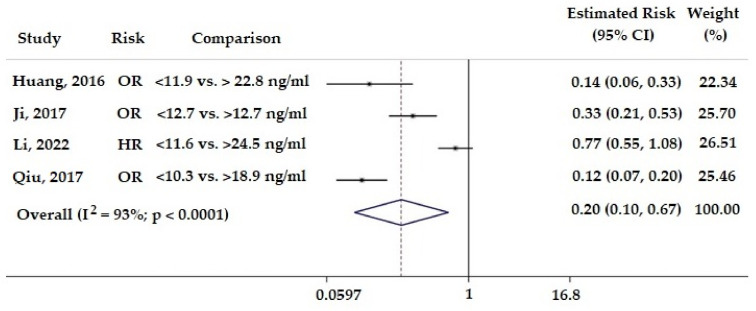
Forest plot of the predicting role of 25(OH)D levels on the risk of recurrent stroke [5,23,24,25].

**Figure 3 nutrients-15-00512-f003:**
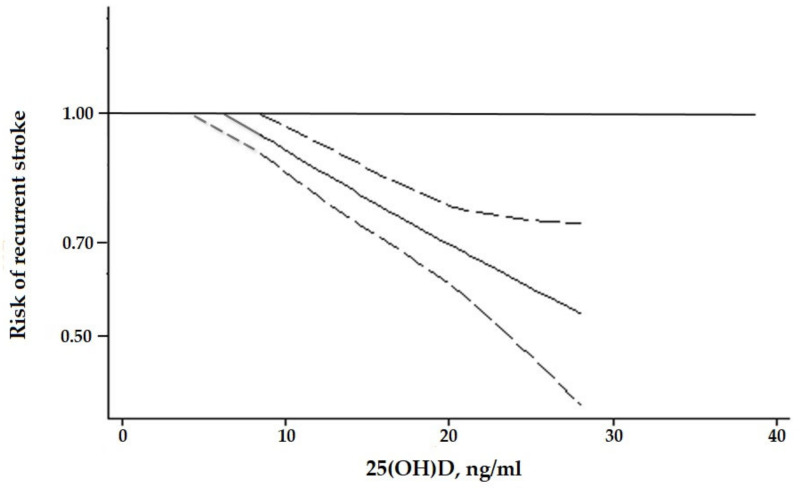
Dose-response association between vitamin D and the risk of stroke recurrence.

**Figure 4 nutrients-15-00512-f004:**
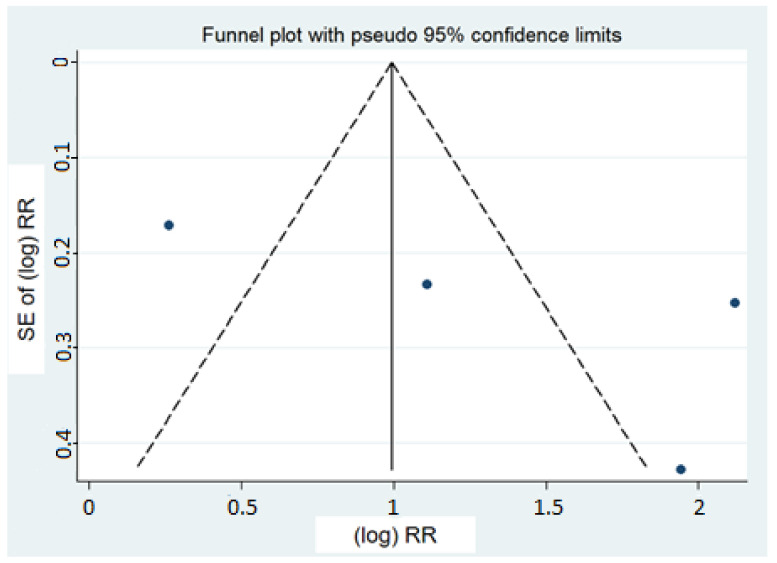
Funnel plot for all of the included studies. SE: standard error; RR: relative risk.

**Table 1 nutrients-15-00512-t001:** Main characteristics of the study populations included in meta-analysis.

Author	Year	Ref	SD	P	M	Age	FU	HTN	T2D	AF	HCL	SM	R-S	Death	NOS
Qiu H	2017	[23]	p	220	61.6	65	24	80.1	43.1	25.0	45.8	25.9	40	30.1	8
Ji W	2017	[25]	p	277	58.9	65	6	59.6	28.2	22.4	29.2	26.4	31	-	7
Huang H	2016	[24]	p	396	58.2	68	3	54.2	32.1	26.1	35.8	18.6	37	11.9	7
Li G	2022	[5]	p	6824	59.2	60.6	86	61.9	14.7	7.8	43.9	16.2	388	9.6	7

Author: name of the first author; Year: year of publication; Ref: reference; SD: study design; P: absolute number of patients; M: percentage of males; Age: mean age expressed in years; FU: duration of follow-up expressed in months; HTN: percentage of subjects affected by hypertension; T2D: percentage of subjects affected by type 2 diabetes; AF: percentage of subjects affected by atrial fibrillation; HCL: percentage of subjects affected by hypercholesterolemia; SM: percentage of smoker subjects; R-S: absolute number of subjects with recurrent stroke; Death: mortality rate; NOS: Newcastle–Ottawa Scale; r: retrospective study.

## Data Availability

Not applicable.

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
