# Peer review of "25-Hydroxy-Vitamin D and Risk of Recurrent Stroke: A Dose Response Meta-Analysis"

_nutrients, 2023, doi:10.3390/nu15030512_

Round 1

Reviewer 1 Report

Good work from the point of view of methodology of meta-analysis and statistical processing with the conclusion "stroke patients with low 25OHD serum levels have an increased risk of recurrence" The meta-analysis includes 4 works, in which the included patients are from the same country (China). The conclusions of the mentioned works are similar, for example: - "Lower serum levels of 25(OH) D are independently associated with the stroke recurrence and mortality at 24 months in ischemic stroke patients" (J Nutr Health Aging . 2017;21(7):766-7710; - "reduced serum levels of 25(OH) D can predict the risk of early stroke recurrence in patients with first-ever ischemic stroke" (Comparative Study Nutr Metab Cardiovasc Dis. 2016 Oct;26(10):908-14); - "Our results demonstrate that low 25(OH) D levels are associated with stroke recurrence and support the hypothesis that 25(OH) D may serve as a biomarker of poor functional outcome after stroke" (J Nutr Health Aging 2017, 21, 892–896).
The question is whether a meta-analysis of the results of few studies is needed to prove what has already been proven.

There is the lack of accuracy: 1) figure 1: Not 4 articles excluded, but 5 articles (last box on the right side) 2) The number of included patients is unclear:
overall 7388 participants were included (line 120) referring to publications no. 17-19, but in the table referring to these same publications there are 7440 patients. In line 133: Four studies and 7554 total patients were included in this analysis [17-20] (Table 1). However, Table 1 shows 7717 patients included

Author Response

Dear Madame/Sir,

We wish to express our appreciation for your comment to our paper. Your suggestions have been considered in preparing the attached revised version of the manuscript with particular attention for the points:

Comments to the Author

Good work from the point of view of methodology of meta-analysis and statistical processing with the conclusion "stroke patients with low 25OHD serum levels have an increased risk of recurrence" The meta-analysis includes 4 works, in which the included patients are from the same country (China). The conclusions of the mentioned works are similar, for example: -"Lower serum levels of 25(OH) D are independently associated with the stroke recurrence and mortality at 24 months in ischemic stroke patients" (J Nutr Health Aging . 2017;21(7):766-7710; - "reduced serum levels of 25(OH) D can predict the risk of early stroke recurrence in patients with first-ever ischemic stroke" (Comparative Study Nutr Metab Cardiovasc Dis. 2016 Oct;26(10):908-14); - "Our results demonstrate that low 25(OH)D levels are associated with stroke recurrence and support the hypothesis that 25(OH) D may serve as a biomarker of poor functional outcome after stroke" (J Nutr Health Aging 2017, 21,892–896). The question is whether a meta-analysis of the results of few studies is needed to prove what has already been proven.

Answer to the Reviewer 1

According to your very interesting comment, we have rewritten some sentences in the manuscript

Our meta-analysis suffers from two major limitations: heterogeneity and a small number of eligible studies, all of them were however performed in the same Country (China). (Line 199-201)

In particular, the dose response analysis is functional to partially overcome the study limitations and to provide new and intriguing data (Line 206-208)

There is the lack of accuracy: 1) figure 1: Not 4 articles excluded, but 5 articles (last box on the right side) 2) The number of included patients is unclear: overall 7388 participants were included (line 120) referring to publications no. 17-19, but in the table referring to these same publications there are 7440 patients. In line 133: Four studies and 7554 total patients were included in this analysis [17-20] (Table 1). However, Table 1 shows 7717 patients included.

Answer to the Reviewer 1

We apologize for the mistakes. According to your suggestion and to those of other reviewers figure 1, table 1, figure 2, and the sentences in the Results section were rewritten

Lines 112- 123: From a total of 71 publications retrieved, 4 prospective studies [18- 21] were identified that met the inclusion criteria (Figure 1). All studies were published between 2016 and 2022. Three studies enrolled patients with just ischemic stroke [18, 20, 21], whereas one study enrolled patients with both ischemic and haemorrhagic events [19]. Overall, 7717 patients were included in this analysis [18- 21] (Table 1). During the follow-up, 496 recurrent events have been documented. Only one study clarifies if the stroke recurrence was ischemic or haemorrhagic [19]. Three retrospective studies were included in the dose-response evaluation of the association between 25(OH)D and risk of recurrent stroke (overall, 7440 participants and 465 recurrent stroke) [18- 20].

Yours sincerely

Anita Vergatti

Lanfranco D’Elia

Domenico Rendina

Reviewer 2 Report

Was the study was only for ischemic stroke. If so, that should be stated in the title and abstract. If not, Hemorrhagic strokes should also be discussed.

From a web search:

What are the chances of having a second hemorrhagic stroke?

Of those, the CDC notes, about 25 percent occur in those who have already suffered a stroke. This includes both ischemic strokes, where a blood clot blocks blood flow to the brain, and hemorrhagic strokes, when an artery in the brain breaks open. “One in four people who have a stroke may have another,” says Dr.May 27, 2022

From a total of 71 publications retrieved, 5 prospective studies [17- 21] were identified that met the inclusion criteria (Figure 1).

Comment: Later #21 was omitted and it was only 4.

The analysis of the four studies is OK. However, the authors are not very familiar with the relevant vitamin D peer-reviewed journal literature regarding related issues. There is reasonable evidence that 25OHD concentrations should be above 30 ng/mL if not 40 ng/mL and that vitamin D reduces risk of several related diseases in a causal mannet.

Vitamin D RCTs have mostly been based on guidelines for pharmaceutical drugs. As a result, most of the participants have high 25OHD concentrations, are given low vitamin D doses, and no account is taken of baseline and achieved 25OHD.

Heaney outlined how they should be based for nutrients, which, in the case of vitamin D, would be 25(OH)D.

Guidelines for optimizing design and analysis of clinical studies of nutrient effects.

Heaney RP.Nutr Rev. 2014 Jan;72(1):48-54. doi: 10.1111/nure.12090. 

Also, well-conducted Mendelian randomization studies are replacing RCTs for vitamin D causality. They use genome-wide association studies (GWAS) to determine genetically-predicted 25(OH)D concentrations for participants in large databases, then do statistical analyses on health outcomes with respect to the genetically-predicted concentrations. Thus, the participants are randomized in a manner that would average out such factors as oral vitamin D intake and solar UVB exposure, providing that enough participants are included.

Vitamin D Deficiency Increases Mortality Risk in the UK Biobank : A Nonlinear Mendelian Randomization Study.

Sutherland JP, Zhou A, Hyppönen E.Ann Intern Med. 2022 Nov;175(11):1552-1559. doi: 10.7326/M21-3324.

Genetic Determinants of 25-Hydroxyvitamin D Concentrations and Their Relevance to Public Health.

Hyppönen E, Vimaleswaran KS, Zhou A.Nutrients. 2022 Oct 20;14(20):4408. doi: 10.3390/nu14204408.

Non-linear Mendelian randomization analyses support a role for vitamin D deficiency in cardiovascular disease risk.

Zhou A, Selvanayagam JB, Hyppönen E.Eur Heart J. 2022 May 7;43(18):1731-1739. doi: 10.1093/eurheartj/ehab809.

Mendelian randomization studies also support the role of vitamin D in reducing risk of  recurrent stroke

Mendelian randomization focused analysis of vitamin D on the secondary prevention of ischemic stroke

YH Chan, CM Schooling, J Zhao, SL Au Yeung, JJ Hai… - Stroke, 2021 - Am Heart Assoc

27. Thomas, M.K.; Lloyd-Jones, D.M.; Thadhani, R.I.; Shaw, A.C.; Deraska, D.J.; Kitch, B.T.; Vamvakas, E.C.; Dick, I.M.; Prince, R.L.; 263 Finkelstein, J.S. Hypovitaminosis D in medical inpatients. N Engl J Med 1998, 338, 777–783. 264

28. Pilz, S.; Tomaschitz, A.; Ritz, E.; Pieber, T.R. Vitamin D status and arterial hypertension: a systematic review. Nat Rev Cardiol 265 2009, 6, 621–630.

See:
Effects of age and serum 25-OH-vitamin D on serum parathyroid hormone levels.

Valcour A, Blocki F, Hawkins DM, Rao SD.J Clin Endocrinol Metab. 2012 Nov;97(11):3989-95. doi: 10.1210/jc.2012-2276. 

The Association between Serum 25(OH)D Status and Blood Pressure in Participants of a Community-Based Program Taking Vitamin D Supplements.

Mirhosseini N, Vatanparast H, Kimball SM.Nutrients. 2017 Nov 14;9(11):1244. doi: 10.3390/nu9111244.

29. Pittas, A.G.; Dawson-Hughes, B. Vitamin D and diabetes. J Steroid Biochem Mol Biol 2010, 121, 425–429.

Comment: poor analysis; See:

Intratrial Exposure to Vitamin D and New-Onset Diabetes Among Adults With Prediabetes: A Secondary Analysis From the Vitamin D and Type 2 Diabetes (D2d) Study.

Dawson-Hughes B, Staten MA, Knowler WC, Nelson J, Vickery EM, LeBlanc ES, Neff LM, Park J, Pittas AG; D2d Research Group.Diabetes Care. 2020 Dec;43(12):2916-2922. doi: 10.2337/dc20-1765.

33. Manson, J.E.; Cook, N.R.; Lee, I.M.; Christen, W.; Bassuk, S.S.; Mora, S.; Gibson, H.; Gordon, D.; Copeland, T.; D'Agostino, D.; 274 Friedenberg, G.; Ridge, C.; Bubes, V.; Giovannucci, E.L.; Willett, W.C.; Buring, J.E.; VITAL Research Group. Vitamin D Supple-275 ments and Prevention of Cancer and Cardiovascular Disease. N Engl J Med 2019, 380, 33–44.

Also poorly designed, conducted and analyzed. Look at the secondary analyses, which showed beneficial effects for cancer.

35. Ross, A.C.; Taylor, C.L.; Yaktine, A.L.; Del Valle, H.B. Institute of Medicine (US) Committee to Review Dietary Reference Intakes 284 for Vitamin D and Calcium. Dietary Reference Intakes for Calcium and Vitamin D 2011, National Academies Press (US). 285

37. Bouillon, R.; Carmeliet, G. Vitamin D insufficiency: Definition, diagnosis and management. Best Pract Res Clin Endocrinol Metab 289 2018, 32, 669–684.

Comment: 30 ng/mL has more support from various studies than does 20 ng/mL.

36. Holick, M.F.; Binkley, N.C.; Bischoff-Ferrari, H.A.; Gordon, C.M.; Hanley, D.A.; Heaney, R.P.; Hassan Murad, M.; Weaver, C.M. 286 Evaluation, Treatment, and Prevention of Vitamin D Deficiency: An Endocrine Society Clinical Practice Guideline. J Clin Endo-287 crinol Metab 2011, Volume 96, pp. 1911–1930. 288

Clinical Practice in the Prevention, Diagnosis and Treatment of Vitamin D Deficiency: A Central and Eastern European Expert Consensus Statement.

Pludowski P, Takacs I, Boyanov M, Belaya Z, Diaconu CC, Mokhort T, Zherdova N, Rasa I, Payer J, Pilz S.Nutrients. 2022 Apr 2;14(7):1483. doi: 10.3390/nu14071483.

Immunologic Effects of Vitamin D on Human Health and Disease.

Charoenngam N, Holick MF.Nutrients. 2020 Jul 15;12(7):2097. doi: 10.3390/nu12072097.

Vitamin D for skeletal and non-skeletal health: What we should know.

Charoenngam N, Shirvani A, Holick MF.J Clin Orthop Trauma. 2019 Nov-Dec;10(6):1082-1093. doi: 10.1016/j.jcot.2019.07.004. 

Isn’t this paper a bit old?

Dusso, A.S.; Brown, A.J.; Slatopolsky, E. Vitamin D.

This article is much better and has been cited over 10,000 times

Vitamin D deficiency.

Holick MF.N Engl J Med. 2007 Jul 19;357(3):266-81. doi: 10.1056/NEJMra070553.

Author Response

Dear Madame/Sir,

We wish to express our appreciation for your comment to our paper. Your suggestions have been considered in preparing the attached revised version of the manuscript with particular attention for the points:

Comments to the Author

Was the study was only for ischemic stroke. If so, that should bestated in the title and abstract. If not, Hemorrhagic strokes should also be discussed

From a web search:

What are the chances of having a second hemorrhagic stroke?

Of those, the CDC notes, about 25 percent occur in those who have already suffered a stroke. This includes both ischemic strokes, where a blood clot blocks blood flow to the brain, and hemorrhagic strokes, when an artery in the brain breaks open. “One in four people who have a stroke may have another,”says Dr.May 27, 2022

Answer to the Reviewer 2

According to your very interesting comments, we have rewritten several sentences in the manuscript

Introduction section: Each year, about 795000 individuals in the United States experience a stroke [2], of whom 87% is represented by ischemic stroke and the rest of haemorrhagic [1, 2]. The most important risk factors for stroke are atherosclerosis, hypertension, vascular abnormalities, diabetes, obesity, and cigarette smoke [2], but the identification of other possible risk factors is still ongoing.  (Line 42-47)

According to several meta-analyses, low 25(OH)D are significantly related with the risk of a first stroke [10-13], but few data are available regarding the relationship between 25(OH)D and risk of recurrent stroke in patients with a first haemorrhagic or ischemic event [14]. (Line 57-60)

Materials and methods section: The included studies satisfy the following criteria: i) prospective or retrospective observational studies enrolling patients with all kind of stroke (Line 71, 72)

Results section: Three studies enrolled patients with just ischemic stroke [18, 20, 21], whereas one study enrolled patients with both ischemic and haemorrhagic events [19]. (Line 116- 118)

During the follow-up, 496 recurrent events have been documented. Only one study clarifies if the stroke recurrence was ischemic or haemorrhagic [19] (Line 119, 120)

From a total of 71 publications retrieved, 5 prospective studies [17- 21] were identified that met the inclusion criteria (Figure 1).

Comment: Later #21 was omitted and it was only 4.

Answer to the Reviewer 2

We apologize for the mistakes. According to your suggestion and to those of other reviewers figure 1 and the sentences in the Results section were rewritten

Lines 112- 123: From a total of 71 publications retrieved, 4 prospective studies [18- 21] were identified that met the inclusion criteria (Figure 1). All studies were published between 2016 and 2022. Three studies enrolled patients with just ischemic stroke [18, 20, 21], whereas one study enrolled patients with both ischemic and haemorrhagic events [19]. Overall, 7717 patients were included in this analysis [18- 21] (Table 1). During the follow-up, 496 recurrent events have been documented. Only one study clarifies if the stroke recurrence was ischemic or haemorrhagic [19]. Three retrospective studies were included in the dose-response evaluation of the association between 25(OH)D and risk of recurrent stroke (overall, 7440 participants and 465 recurrent stroke) [18- 20].

The analysis of the four studies is OK.

Thank you

However, the authors are not very familiar with the relevant vitamin D peer-reviewed journal literature regarding related issues. There is reasonable evidence that 25OHD concentrations should be above 30 ng/mL if not 40 ng/mL and that vitamin D reduces risk of several related diseases in a causal mannet.

According to your suggestions, we have added this sentence in the Discussion section:

]. On the opposite, there is reasonable evidence that 25(OH)D concentration should be above 30 ng/ml to reduce the risk of several, multifactorial, and complex disease [49] (Line 195- 197).

Vitamin D RCTs have mostly been based on guidelines for pharmaceutical drugs. As a result, most of the participants have high 25OHD concentrations, are given low vitamin D doses, and no account is taken of baseline and achieved 25OHD.

According to your suggestions, we have added this sentence in the Discussion section:

In this regard, it should be noted that most of the participant to large vitamin D randomized controlled trials have high 25(OH)D concentrations and no account is taken of baseline 25(OH)D [43, 44] (Line 186- 188)

Heaney outlined how they should be based for nutrients, which, in the case of vitamin D, would be 25(OH)D.

Guidelines for optimizing design and analysis of clinical studies of nutrient effects. Heaney RP. Nutr Rev. 2014 Jan;72(1):48-54. doi:10.1111/nure.12090

Also, well-conducted Mendelian randomization studies are replacing RCTs for vitamin D causality. They use genome-wide association studies (GWAS) to determine genetically-predicted 25(OH)D concentrations for participants in large databases, then do statistical analyses on health outcomes with respect to the genetically-predicted concentrations. Thus, the participants are randomized in a manner that would average out such factors as oral vitamin D intake and solar UVB exposure, providing that enough participants are included.

According to your suggestion, we have added this sentence in Discussion section:

Furthermore, the studies included in this analysis take accounts just of 25(OH)D levels, without considering nutritional supplementation and solar ultra-violet B exposure, outperformed by genetic mendelian randomized studies [50- 53] (Line 202- 205).

Nevertheless, our results are supported by genetic mendelian randomized studies [50]. (Line 197, 198)

In addition, vitamin D serum levels have been abbreviated according to Heaney’s indications.

  • Vitamin D Deficiency Increases Mortality Risk in the UK Biobank: A Nonlinear Mendelian Randomization Study. Sutherland JP, Zhou A, Hyppönen E.Ann Intern Med. 2022Nov;175(11):1552-1559. doi: 10.7326/M21-3324
  • Genetic Determinants of 25-Hydroxyvitamin D Concentrationsand Their Relevance to Public Health. Hyppönen E, Vimaleswaran KS, Zhou A. Nutrients. 2022 Oct20;14(20):4408. doi: 10.3390/nu14204408.
  • Zhou A, Selvanayagam JB, Hyppönen E.Eur Heart J. 2022 May 7;43(18):1731-1739. doi: 10.1093/eurheartj/ehab809.
  • Mendelian randomization focused analysis of vitamin D on the secondary prevention of ischemic stroke YH Chan, CM Schooling, J Zhao, SL Au Yeung, JJHai… - Stroke, 2021 - Am Heart Assoc
  • Valcour A, Blocki F, Hawkins DM, Rao SD .J Clin EndocrinolMetab. 2012 Nov;97(11):3989-95. doi: 10.1210/jc.2012-2276.
  • The Association between Serum 25(OH)D Status and Blood Pressure in Participants of a Community-Based Program Taking Vitamin D Supplements.Mirhosseini N, Vatanparast H, Kimball SM.Nutrients. 2017 Nov14;9(11):1244. doi: 10.3390/nu9111244
  • Intratrial Exposure to Vitamin D and New-Onset Diabetes Among Adults With Prediabetes: A Secondary Analysis From the Vitamin D and Type 2 Diabetes (D2d) Study. Dawson-Hughes B, Staten MA, Knowler WC, Nelson J, VickeryEM, LeBlanc ES, Neff LM, Park J, Pittas AG; D2d ResearchGroup. Diabetes Care. 2020 Dec;43(12):2916-2922. doi:10.2337/dc20-1765
  • Clinical Practice in the Prevention, Diagnosis and Treatmentof Vitamin D Deficiency: A Central and Eastern European Expert Consensus Statement. Pludowski P, Takacs I, Boyanov M, Belaya Z, Diaconu CC,Mokhort T, Zherdova N, Rasa I, Payer J, Pilz S.Nutrients. 2022Apr 2;14(7):1483. doi: 10.3390/nu14071483
  • Immunologic Effects of Vitamin D on Human Health andDisease. Charoenngam N, Holick MF. Nutrients. 2020 Jul 15;12(7):2097.doi: 10.3390/nu12072097.
  • Vitamin D for skeletal and non-skeletal health: What we should know. Charoenngam N, Shirvani A, Holick MF. J Clin Orthop Trauma.2019 Nov-Dec;10(6):1082-1093. doi:10.1016/j.jcot.2019.07.004.
  •  Vitamin D deficiency. Holick MF. N Engl J Med. 2007 Jul 19;357(3):266-81. doi:10.1056/NEJMra070553.

According to your suggestion, all these articles were added to the reference list and discussed

  1. Manson, J.E.; Cook, N.R.; Lee, I.M.; Christen, W.; Bassuk,S.S.; Mora, S.; Gibson, H.; Gordon, D.; Copeland, T.; D'Agostino,D.; 274 Friedenberg, G.; Ridge, C.; Bubes, V.; Giovannucci, E.L.;Willett, W.C.; Buring, J.E.; VITAL Research Group. Vitamin D Supplements and Prevention of Cancer and Cardiovascular Disease. N Engl J Med 2019, 380, 33–44.

Also poorly designed, conducted and analyzed. Look at the secondary analyses, which showed beneficial effects for cancer.  

Thanks for your suggestion. We have added this sentence in the Discussion section:

the results of VITAL study showed that indiscriminate supplementation of vitamin D is not useful for the primary prevention of cardiovascular diseases, although significantly improves the prognosis quoad vitam et valitudinem of patients with cancer [42] (Line 183-186)

  1. Bouillon, R.; Carmeliet, G. Vitamin D insufficiency: Definition, diagnosis and management. Best Pract Res Clin EndocrinolMetab 289 2018, 32, 669–684.

Comment: 30 ng/mL has more support from various studies than does 20 ng/mL.

According to your suggestion, this sentence is added in Discussion section:

]. On the opposite, there is reasonable evidence that 25(OH)D concentration should be above 30 ng/ml to reduce the risk of several, multifactorial, and complex disease [49] (Line 195-197)

Yours sincerely

Anita Vergatti

Lanfranco D’Elia

Domenico Rendina

Reviewer 3 Report

The reviewer appreciate the interest of the authors in the development of this manuscript. It is an interesting topic.

In their paper, the authors reviewed the literature on the association between serum 25-hydroxy-D vitamin levels and the risk of recurrent stroke.  The manuscript is generally well written. However, before this manuscript is suitable for publication, I propose a small correction:

1.       Increase the font in figure 2

The reviewer appreciate the interest of the authors in the development of this manuscript. It is an interesting topic.

Author Response

Dear Madame/Sir,

We wish to express our appreciation for your comment to our paper. Your suggestion has been considered in preparing the attached revised version of the manuscript with particular attention for the points:

Comments to the Author

The reviewer appreciate the interest of the authors in the development of this manuscript. It is an interesting topic.

In their paper, the authors reviewed the literature on the association between serum 25-hydroxy-D vitamin levels and the risk of recurrent stroke. The manuscript is generally well written. However, before this manuscript is suitable for publication, I propose a small correction:

  1. Increase the font in figure 2

The reviewer appreciate the interest of the authors in the development of this manuscript. It is an interesting topic.

Answer to the Reviewer 3

Thanks for your comment. We have increased the font in figure 2 and modified the figure footnotes accordingly.

Yours sincerely

Anita Vergatti

Lanfranco D’Elia

Domenico Rendina

Round 2

Reviewer 1 Report

My objections have been removed. Can be published despite so-so merit

Reviewer 2 Report

The manuscript has been considerably improved.

No furher comments.